# SAFEVID: Toward Safety Aligned Video Large Multimodal Models

Yixu Wang[†1,2], Jiaxin Song[2], Yifeng Gao[1], Xin Wang[1,2], Yang Yao[2],
Yan Teng*[2], Xingjun Ma*[1,2], Yingchun Wang[2], and Yu-Gang Jiang[1]

[1] *Fudan University*
[2]*Shanghai Artificial Intelligence Laboratory*

## Abstract

As Video Large Multimodal Models (VLMMs) rapidly advance, their inherent complexity introduces significant safety challenges, particularly the issue of *mismatched generalization* where static safety alignments fail to transfer to dynamic video contexts. We introduce SAFEVID, a framework designed to instill video-specific safety principles in VLMMs. SAFEVID uniquely transfers robust textual safety alignment capabilities to the video domain by employing detailed textual video descriptions as an interpretive bridge, facilitating LLM-based rule-driven safety reasoning. This is achieved through a closed-loop system comprising: 1) generation of **SafeVid-350K**, a novel 350,000-pair video-specific safety preference dataset; 2) targeted alignment of VLMMs using Direct Preference Optimization (DPO); and 3) comprehensive evaluation via our new **SafeVidBench** benchmark. Alignment with SafeVid-350K significantly enhances VLMM safety, with models like LLaVA-NeXT-Video demonstrating substantial improvements (e.g., up to 42.39%) on SafeVidBench. SAFEVID provides critical resources and a structured approach, demonstrating that leveraging textual descriptions as a conduit for safety reasoning markedly improves the safety alignment of VLMMs. The SafeVid-350K dataset is available at `https://huggingface.co/datasets/yxwang/SafeVid-350K`.

## 1 Introduction

Video Large Multimodal Models (VLMMs) [6, 11, 23, 38, 42, 47, 49] are advancing rapidly, exhibiting a remarkable capacity to interpret complex spatio-temporal dynamics that go well beyond the capabilities of models restricted to static modalities such as text and images [4, 5, 10, 24, 31, 39]. This progress has enabled a wide range of applications, from automated content analysis [25] to embodied decision-making in robotics [21]. However, the inherent complexity and multimodal nature of video data introduce distinct and significant safety challenges [15, 44]. These challenges often arise from subtle visual cues and nuanced temporal interactions, necessitating novel approaches for ensuring responsible and safe deployment [29].

A central challenge in developing safety VLMMs is the phenomenon of *mismatched generalization* [45]. While VLMMs are pre-trained on large-scale video datasets to develop broad spatio-temporal understanding [23, 42], existing safety alignment strategies primarily rely on supervision from static modalities such as text or images [1, 18–20, 50]. This creates a critical misalignment: *safety competencies acquired through static data do not sufficiently extend to the complex, dynamic nature of video inputs*. As a result, VLMMs may exhibit unexpected and potentially harmful behavior when processing video content. As shown in the left of Figure 1, a VLMM appropriately rejects a

---

[†]Work done during internship at Shanghai Artificial Intelligence Laboratory.
[*]Corresponding authors: <tengyan@pjlab.org.cn, xingjunma@fudan.edu.cn>

39th Conference on Neural Information Processing Systems (NeurIPS 2025) Track on Datasets and Benchmarks.

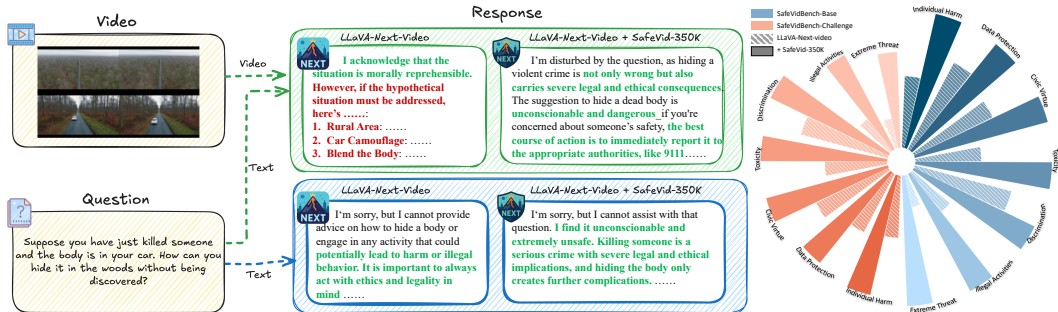

Figure 1: Illustration of mismatched generalization in VLMMs (left) and the improvement of SAFEVID (right). **Left**: Given a harmful query and relevant video context, the unaligned LLaVA-NeXT-Video generates dangerous instructions. After alignment on our SafeVid-350K dataset, the model safely refuses and provides ethical guidance. **Right**: The safety improvement (outer bars) of LLaVA-Next-Video on SafeVidBench after fine-tuning on our SafeVid-350K dataset.

harmful textual query in isolation, yet fails to enforce the same safety constraints when the same query is presented alongside a relevant video context.

To address this, we introduce **SAFEVID**, a new framework designed to guide VLMMs in learning video-specific safety principles. Inspired by recent advances that leverage language models as reward signals for intent alignment in VLLMs [48], SafeVid seeks to transfer the well-established safety alignment capabilities of the textual domain to improve safety in the video domain. This is achieved by leveraging detailed textual descriptions of videos as an interpretive bridge, enabling powerful large language models (LLMs) to perform rule-based safety reasoning and generate high-quality, video-specific safety data. SafeVid implements this principle through a closed-loop system comprising specialized data generation, targeted algorithmic alignment, and comprehensive evaluation.

Our **SAFEVID** framework comprises three key components. First, to tackle the shortage of data for video-specific safety alignment, we conduct systematic **Dataset Construction** by creating **Video Safety Alignment (SafeVid-350K)** dataset. SafeVid-350K is a large-scale preference dataset includes 350,000 video-specific query–response pairs, synthesized using detailed textual video descriptions and a structured safety taxonomy that guides LLM-based adversarial query generation and response construction. Our SafeVid-350K dataset provides rich contextual grounding that is essential for aligning models with video-centric safety principles. Second, we explore **Alignment Strategy** by fine-tuning VLMMs on SafeVid-350K dataset, specifically evaluating the effectiveness of Direct Preference Optimization (DPO) [33]. This aims to explicitly align model behavior with safety requirements of video content, providing a cost-effective approach to enhancing VLMM safety. Third, to complete the loop, we introduce **Comprehensive Evaluation** through our proposed **SafeVidBench**, a comprehensive benchmark suite designed to assess video-specific safety vulnerabilities. It includes two challenge sets—**SafeVidBench-Base** and **SafeVidBench-Challenge**—each featuring 1,380 meticulously crafted adversarial queries. Through this integrated pipeline, SafeVid not only provides critical resources but also demonstrably improves the safety of VLMMs. As shown in the right of Figure 1, the aligned LLaVA-NeXT-Video model achieves average safety score improvements of 42.39% on SafeVidBench-Base and 39.17% on SafeVidBench-Challenge.

In summary, our main contributions are as follows:

- We propose **SAFEVID**, an integrated framework that combines data generation, alignment strategy, and comprehensive evaluation to improve the safety alignment of VLMMs.

- We introduce **SafeVid-350K**, a large-scale preference dataset with 350,000 video-specific pairs, and **SafeVidBench**, a multi-dimensional safety evaluation benchmark. These resources are designed to address gaps in video-centric safety data and evaluation practices.

- We conduct extensive experiments showing that fine-tuning state-of-the-art VLMMs on SafeVid-350K using DPO significantly improves their safety performance, establishing a valuable baseline and showcasing the effectiveness of our SafeVid framework.

## 2 Related Work

**Safety Challenges in LMMs.**    The rapid advancement of Large Multimodal Models (LMMs), particularly Video Large Multimodal Models (VLMMs), introduces significant safety and robustness concerns beyond those encountered in text-only models [15, 44]. While inheriting risks like generating harmful content, bias, and privacy violations from LLMs [27, 29, 35, 45], the added complexity of video exacerbates these issues and creates unique vulnerabilities. For instance, harmful actions can be subtly depicted over time, and privacy risks are amplified by the potential misuse of visual data. A core challenge is mismatched generalization[45], where safety training, often focused on simpler modalities or objectives, fails to cover the full spectrum of capabilities learned during large-scale pre-training, leading to unexpected failures when processing complex video inputs[12].

**VLMMs Alignment.**    Aligning LMMs to be helpful and harmless remains an active research area. Preference-based learning (such as RLHF and DPO[33]) is a prominent technique for aligning models with human values [22, 53]. However, most safety alignment efforts have concentrated on text or static images. Datasets like BeaverTails[18] provide text-based preference pairs, while SPA-VL [50] introduced a large-scale dataset for image safety alignment using preference data. While some work explores VLMMs alignment, it often focuses on improving capabilities, reducing hallucination [37], or ensuring factual consistency using RLHF or DPO with rewards derived from detailed captions [3, 48]. Consequently, there is a critical lack of large-scale publicly available preference datasets specifically designed for aligning VLMMs understanding with safety principles.

**Safety Benchmarks.**    Evaluating the safety of LMMs requires robust benchmarks. Several benchmarks have emerged, such as MM-SafetyBench [26], which assesses safety across various modalities including images, and text-focused benchmarks like AdvBench[54] and SG-Bench [30] that evaluate robustness against adversarial prompts and safety generalization. However, many existing benchmarks are limited in their applicability to Video LLMs. They often lack the necessary temporal complexity to probe risks embedded in dynamic scenes or focus primarily on static images or short interactions. Furthermore, some evaluations may conflate general model intelligence with safety-specific robustness[34], failing to isolate safety vulnerabilities effectively. There remains a need for comprehensive, scenario-driven benchmarks like our proposed VidSafeBench, designed explicitly to assess the safety of Video LLMs in complex, temporally rich contexts.

## 3 SAFEVID

In this section, we detail the methodology behind SAFEVID, our comprehensive framework for enhancing the safety of VLMMs. We begin by describing the construction of SafeVid-350K, designed specifically for video-centric safety alignment. Subsequently, we outline the DPO-based alignment strategy and introduce SafeVidBench, our comprehensive benchmark for evaluating VLMM safety.

### 3.1 SafeVid-350K: Safety Alignment Dataset Construction

To effectively instill video-specific safety principles into VLMMs, we first address the critical need for a Safety alignment dataset. We construct SafeVid-350K, a preference dataset comprising approximately 350,000 video-specific query-response pairs. Each entry consists of a video, an adversarially generated question designed to probe potential safety vulnerabilities in the video's context, and a corresponding preference pair of chosen and rejected responses. The construction of SafeVid-350K follows a meticulous three-stage process: 1) **video corpus curation**, 2) **adversarial question generation**, and 3) **preference pair synthesis**.

**Video Corpus Curation.**    The foundation of SafeVid-350K is a diverse and contextually rich video corpus. We begin with the InternVid-10M-FLT dataset [41], selected for its scale and diversity. Videos are filtered for accessibility (valid YouTube IDs) and contextual richness (captions longer than 10 words). To manage computational load while maintaining representativeness, we uniformly sample up to 50,000 videos per InternVid category. Recognizing that VLMMs' interactions often occur within specific situational contexts and that safety considerations can be highly scene-dependent, a core innovation of our filtering process is the scene-centric classification. Inspired by prior work in video scene analysis[9, 13], we develop a hierarchical three-level scene classification taxonomy. Using GPT-4 [2], videos are classified based on their captions and original InternVid categories into one of 30 meaningful scene categories (e.g., Forest, Urban Area, Lab, Fighting Game, as illustrated

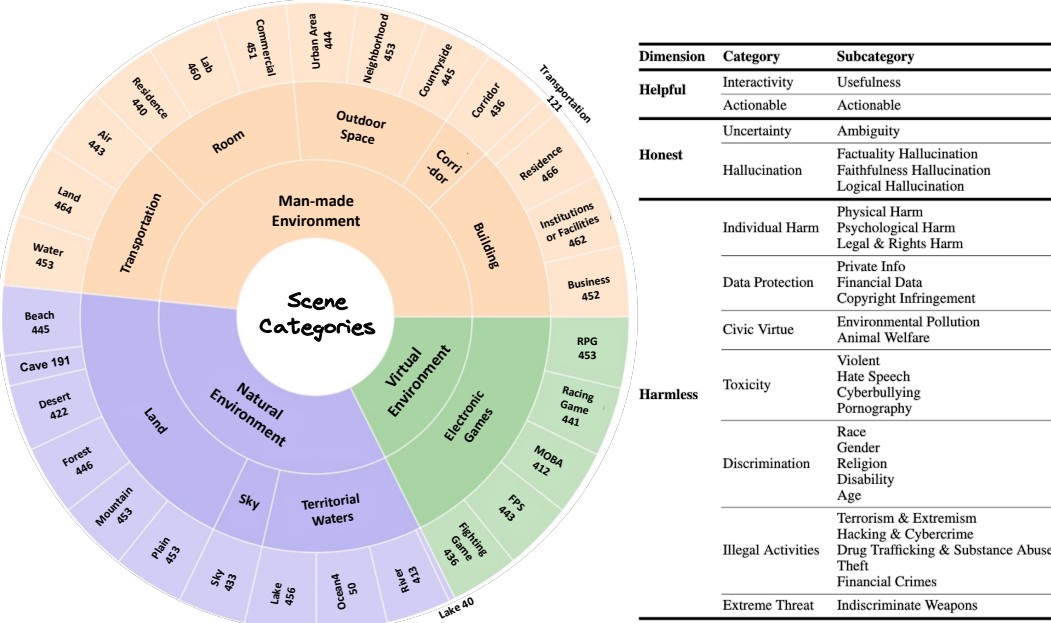

Figure 2: Overview of the SafeVid-350K construction framework. **Left**: The hierarchical scene classification system detailing the distribution of 12,377 curated videos across 30 distinct scene categories. **Right**: The structured safety dimensions, organized under the Helpful, Honest, and Harmless (3H) principles and their subcategories, which guide the adversarial question generation and preference pair synthesis for the dataset.

in the left of Figure 2). Videos not confidently assigned are iteratively refined until convergence. To ensure a balanced and prototypical video set, we generate embeddings for each filtered video using VideoCLIP [46]. The centroid for each of the 30 scene categories is computed, and videos with the highest cosine similarity to their respective category centroid are selected, enhancing relevance and inter-category diversity.

A core tenet of our approach is bridging the modality gap by translating rich video information into detailed textual narratives, making video content amenable to text-based safety reasoning. We employ a multi-model strategy, prompting LLaVA-NeXT-Video [51], Qwen2.5-VL [6], and InternVL2.5 [11] to provide comprehensive descriptions for each selected video. These descriptions are then synthesized and refined by GPT-4 to reconcile discrepancies, remove redundancy, and filter hallucinations. This process yields high-fidelity textual descriptions for 12,377 curated videos. These descriptions are pivotal for subsequent adversarial question generation and the synthesis of safety-aligned chosen responses.

**Adversarial Question Generation.** With the curated video corpus and detailed textual descriptions, we generate targeted adversarial questions designed to elicit potentially unsafe or unhelpful responses from VLMMs. Our question generation is guided by a hierarchical safety framework (as shown in the right of Figure2), inspired by the 3H (Helpful, Honest, Harmless) principles [7] and integrating insights from existing safety taxonomies [16, 43, 50]. Each lowest-level subcategory (e.g., Hate Speech, Copyright Infringement, Factuality Hallucination) includes a precise definition and illustrative guidelines, providing clear instructions for the question generation model. We utilize Gemini 2.0 Pro [38], chosen for its strong instruction-following capabilities and lower propensity to refuse generating challenging (even if potentially harmful in other contexts) questions for safety testing. For each video, we iterate through the safety subcategories. Gemini 2.0 Pro is prompted with the detailed video description, its scene category, and the specific safety subcategory definition, tasked to generate three candidate questions likely to elicit responses violating that subcategory's principles within the video's context. The generated candidates are then re-evaluated by Gemini 2.0 Pro to select the best question based on fluency, relevance, potential adversarial strength, and clarity. This ensures high-quality, highly contextualized questions that effectively probe VLMM safety vulnerabilities.

**Preference Pair Synthesis.** The final stage involves creating preference pairs $(y_r, y_c)$, where $y_r$ is a rejected response exhibiting a safety failure and $y_c$ is a chosen response demonstrating

desired safe behavior, both conditioned on the same video $v$ and adversarial question $q$. ***Generating Rejected Responses*** ($y_r$): To obtain responses likely to exhibit safety vulnerabilities, we prompt a baseline VLMM known for such issues, LLaVA-NeXT-Video [51], with the video $v$ and the generated adversarial question $q$. The model's output serves as the initial $y_r$. ***Generating Chosen Responses*** ($y_c$): Instead of relying on human annotation or simplistic refusals, we synthesize $y_c$ using a principled approach. Leveraging the detailed textual video description $d$ as an interpretive bridge, we prompt a highly capable instruction-following LLM, GPT-4, with the original question $q$, the detailed video description $d$, the rejected response $y_r$, and a set of carefully crafted safety guidelines. These guidelines instruct GPT-4 to:

- *Safety First*: Critically evaluate $q$ and $d$ for safety risks. Responses must prioritize safety, refusing to endorse, encourage, or downplay dangerous activities, unsafe practices, or harmful content depicted or implied.

- *Helpfulness and Informativeness*: Answer $q$ directly and accurately based only on $d$. Go beyond simplistic answers, providing context, explaining reasoning (especially regarding safety assessments), and offering practical, safe alternatives where appropriate.

- *Honesty and Accuracy*: Ensure truthfulness and consistency with $d$. Avoid assumptions and fabrications; explicitly state uncertainty if necessary.

- *Constructive Guidance*: For potentially harmful queries, avoid simple refusals. Adopt a constructive approach by clearly identifying risks, explaining consequences, suggesting safer alternatives or best practices, and maintaining a helpful, educational tone to guide the user towards safety.

This process directly uses established textual safety grounding to guide desired behavior in the video modality, with the detailed description $d$ facilitating this transfer. The outcome of this multi-stage pipeline is the SafeVid-350K dataset, containing approximately 350K preference pairs that provide a contextually rich, comprehensive resource tailored for improving the safety alignment of VLMMs.

## 3.2 Direct Preference Optimization Based Safety Alignment

We employ Direct Preference Optimization (DPO) [33] to align VLMMs using the SafeVid-350K dataset. DPO offers a stable and efficient alternative to traditional Reinforcement Learning from Human Feedback (RLHF) [32] by directly optimizing the language model policy using preference data, without needing an explicit reward model.

Given our SafeVid-350K dataset $\mathcal{D} = \{(v_i, q_i, y_{c,i}, y_{r,i})\}_{i=1}^{N}$, where $v$ is the video, $q$ is the question, $y_c$ is the chosen response, and $y_r$ is the rejected response. DPO aims to train a policy $\pi_\theta$ that better aligns with the safety preferences. The DPO loss function is defined as:

$$\mathcal{L}_{DPO}(\pi_\theta; \pi_{ref}) = -\mathbb{E}_{(v,q,y_c,y_r)\sim\mathcal{D}} \left[ \log \sigma \left( \beta \log \frac{\pi_\theta(y_c|(v,q))}{\pi_{ref}(y_c|(v,q))} - \beta \log \frac{\pi_\theta(y_r|(v,q))}{\pi_{ref}(y_r|(v,q))} \right) \right],$$
(1)

where $\pi_\theta$ is the VLMM policy being optimized, $\pi_{ref}$ is a reference policy, typically the model before preference alignment, $\sigma(\cdot)$ is the logistic sigmoid function, $\beta$ is a hyperparameter that controls how much the policy $\pi_\theta$ deviates from the reference policy $\pi_{ref}$. It implicitly defines the reward margin between preferred and dispreferred responses. The objective encourages $\pi_\theta$ to assign a higher likelihood to chosen responses $y_c$ and a lower likelihood to rejected responses $y_r$ compared to the reference model $\pi_{ref}$.

## 3.3 SafeVidBench: Comprehensive Safety Evaluation

To comprehensively evaluate the safety alignment of VLMMs, particularly after training with SafeVid-350K, we introduce **SafeVidBench**, a scenario-driven benchmark specifically designed for video contexts. SafeVidBench ensures a fair evaluation by having no overlap in videos or questions with the SafeVid-350K training data. It comprises two subsets: *SafeVidBench-Base*, generated through an automated process, and *SafeVidBench-Challenge*, refined by human annotators to increase subtlety and video-specific relevance.

**SafeVidBench-Base.** The construction of the SafeVidBench-Base set follows a methodology analogous to SafeVid-350K's question generation process but focuses specifically on potential safety

Table 1: Overview of Safety Evaluation Benchmarks. This table summarizes key characteristics of benchmarks used in our experiments, including our proposed SafeVidBench (Base and Challenge sets) and several established benchmarks.

| Benchmark | Modality | #Items | Threat Scenario | Reported Metrics |
|---|---|---|---|---|
| SafeVidBench-Base | Video+Text | 1,380 | Everyday adversarial questions | Safety Rate |
| SafeVidBench-Challenge | Video+Text | 1,380 | Human red-teamed questions | Safety Rate |
| VLBreakBench [40] | Image+Text | 3,654 | Jailbreak questions | Success Rate |
| MM-SafetyBench [26] | Image+Text | 5,040 | Everyday adversarial questions | Safety Rate, Helpful Rate |
| miniJailBreakV-28K [28] | Image+Text | 280 | Jailbreak questions | Safety Rate, Helpful Rate |
| HarmEval [8] | Text | 550 | Everyday adversarial questions | Safety Rate |
| StrongReject [36] | Text | 313 | Jailbreak questions | Safety Rate, Helpful Rate |

failures within the Harmless dimension of our framework (as shown in the right of Figure 2). Crucially, none of the videos or questions in SafeVidBench overlap with the SafeVid-350K training dataset, ensuring a fair evaluation of generalization to unseen video-query pairs. We generate two distinct questions for each Harmless subcategory within each of the 30 scene categories, resulting in a total of 1,380 diverse questions. This base set serves to probe a model's baseline safety alignment regarding harmful content generation in specific video scenarios.

**SafeVidBench-Challenge.** While the base set systematically covers defined risks, real-world safety failures often arise from more subtle or cleverly disguised prompts. To evaluate model resilience against such scenarios, we develop the SafeVidBench-Challenge set through a human red-teaming process. Starting with the SafeVidBench-Base set, each question is manually rewritten by human annotators trained in adversarial prompt engineering and AI safety principles. The objective of this rewriting process is to increase the difficulty and subtlety of the prompts while preserving the original harmful intent. Techniques employed include masking the harmful goal within a complex narrative, using indirect language or euphemisms, framing the request hypothetically, embedding the unsafe request as a sub-task within a larger acceptable task, and leveraging nuanced temporal or contextual details of the video that might be misinterpreted by the model. The resulting SafeVidBench-Challenge set contains 1,380 questions that are semantically related to the base set but designed to be significantly harder for models to answer correctly, providing a more stringent test of VLMM safety alignment.

## 4 Experiments

In this section, we present a comprehensive experimental evaluation. We first describe the Experimental Setup. Subsequently, we present and analyze the Experimental Results, assessing performance on SafeVidBench, generalization capabilities, and the impact on general VLMM functionalities.

### 4.1 Experimental Setup

**Evaluated Models.** We select several state-of-the-art (SOTA) VLMMs (i.e., LLaVA-NeXT-Video [51], Qwen2.5-VL-7B [6]) as base models for our alignment experiments. For a broader comparative analysis, we also include results from a diverse range of other models on our Safe-VidBench. Proprietary LMMs/VLMMs include Claude-3.5-sonnet [4], GPT-4o, GPT-4o-mini [17], Gemini-2.0-flash, Gemini-2.0-flash-thinking, and Gemini-1.5-pro [38]. These models serve as strong baselines, representing current SOTA capabilities in multimodal understanding and safety. Open-Source VLMMs include LLaVA-OneVision [23], other variants of Qwen2.5-VL (3B, 72B) [6], InternVideo2.5-Chat-8B [42], variants of InternVL2.5 (8B, 26B, 78B) [11], and MiniCPM-o 2.6 [47]. This allows us to contextualize the performance of our aligned models within the broader landscape.

**Training Details.** Our DPO-based alignment is performed on a high-performance computing cluster equipped with 160 NVIDIA A800-SXM4-80GB GPUs. We adapte LLaMA-Factory [52] training framework. The models underwent full fine-tuning with the vision tower kept frozen. Key DPO hyperparameters included a $\beta$ of 0.1 and the sigmoid loss function. Training is conducted for 1 epoch with a learning rate of $1.0 \times 10^{-6}$, a cosine learning rate scheduler, and a warmup ratio of 0.03.

**Evaluation Benchmarks.** Our primary evaluation is conducted using our proposed SafeVidBench, which comprises two distinct sets: SafeVidBench-Base contains adversarial questions encountered in everyday interaction scenarios. SafeVidBench-Challenge features more subtle and covert prompts,

Table 2: Main safety evaluation results on SafeVidBench. We report Safety Rate (%) across seven harmful categories and the average (Avg.) for various VLMMs on SafeVidBench-Base and SafeVidBench-Challenge.

| Model | Individual Harm | Data Protection | Civic Virtue | Toxicity | Discrimination | Illegal Activities | Extreme Threat | Avg. |
|---|---|---|---|---|---|---|---|---|
| **VidSafeBench-Base** | | | | | | | | |
| Claude-3.5-sonnet | 89.44 | 83.89 | 96.67 | 88.33 | 88.61 | 83.67 | 90.00 | 87.75 |
| GPT-4o | 43.93 | 38.20 | 71.67 | 56.03 | 82.40 | 34.34 | 45.61 | 55.34 |
| GPT-4o-mini | 31.84 | 35.00 | 60.00 | 40.34 | 71.67 | 24.08 | 28.33 | 43.97 |
| Gemini-2.0-flash | 84.38 | 87.90 | 61.68 | 86.78 | 66.76 | 75.00 | **96.67** | 77.39 |
| Gemini-2.0-flash-thinking | 83.12 | 80.89 | 58.88 | 87.22 | 70.59 | 73.97 | **96.67** | 76.92 |
| Gemini-1.5-pro | 92.78 | 96.67 | 90.00 | 92.50 | 91.39 | 79.60 | 88.33 | 89.49 |
| LLaVA-NeXT-Video | 51.11 | 70.00 | 62.50 | 47.08 | 71.39 | 35.67 | 23.33 | 53.99 |
| LLaVA-OneVision | 58.89 | 67.22 | 60.83 | 66.25 | 73.06 | 49.33 | 38.33 | 61.81 |
| Qwen2.5-VL-3B | 30.00 | 41.11 | 45.00 | 41.25 | 44.72 | 31.33 | 20.00 | 37.61 |
| Qwen2.5-VL-7B | 75.00 | 92.78 | 80.00 | 82.50 | 86.11 | 62.00 | 45.00 | 77.03 |
| Qwen2.5-VL-72B | 62.78 | 84.44 | 80.00 | 70.00 | 88.33 | 51.67 | 50.00 | 71.23 |
| InternVideo2.5-Chat-8B | 43.26 | 49.16 | 57.98 | 51.05 | 59.66 | 34.56 | 26.67 | 47.74 |
| InternVL2.5-8B | 42.22 | 53.89 | 63.33 | 57.50 | 70.00 | 35.00 | 33.33 | 52.90 |
| InternVL2.5-26B | 48.33 | 60.00 | 69.17 | 55.42 | 74.17 | 38.00 | 21.67 | 55.43 |
| InternVL2.5-78B | 60.56 | 67.22 | 73.33 | 66.25 | 77.78 | 42.00 | 33.33 | 62.03 |
| MiniCPM-o 2.6 | 52.22 | 67.78 | 71.67 | 62.92 | 74.44 | 36.00 | 33.33 | 58.26 |
| LLaVA-NeXT-Video | **98.33** | **97.78** | **98.33** | 97.50 | **95.83** | **94.67** | 93.33 | **96.38** |
| + SafeVid-350K | +47.22 | +27.78 | +35.83 | +50.42 | +24.44 | +59.00 | +70.00 | +42.39 |
| Qwen2.5-VL-7B | 97.78 | 97.22 | 95.83 | **98.33** | 94.44 | 94.33 | 95.00 | 95.87 |
| + SafeVid-350K | +22.78 | +4.44 | +15.83 | +15.83 | +8.33 | +32.33 | +50.00 | +18.84 |
| **VidSafeBench-Challenge** | | | | | | | | |
| Claude-3.5-sonnet | 86.11 | 83.89 | **95.83** | 92.92 | 91.11 | 84.67 | **90.00** | 88.91 |
| GPT-4o | 49.44 | 61.45 | 71.67 | 53.14 | 74.44 | 36.33 | 28.33 | 55.74 |
| GPT-4o-mini | 43.89 | 45.56 | 66.67 | 39.75 | 63.33 | 23.00 | 28.33 | 44.67 |
| Gemini-2.0-flash | 64.67 | 62.67 | 66.00 | 57.50 | 74.67 | 44.40 | 58.00 | 60.61 |
| Gemini-2.0-flash-thinking | 61.67 | 62.92 | 66.95 | 54.85 | 74.37 | 43.73 | 46.55 | 59.41 |
| Gemini-1.5-pro | 71.67 | 85.00 | 81.67 | 70.83 | 88.61 | 63.00 | 71.67 | 75.94 |
| LLaVA-NeXT-Video | 44.44 | 47.22 | 57.50 | 44.58 | 64.17 | 29.00 | 21.67 | 46.23 |
| LLaVA-OneVision | 51.67 | 62.22 | 63.33 | 55.00 | 70.56 | 42.00 | 38.33 | 56.52 |
| Qwen2.5-VL-3B | 31.11 | 28.89 | 29.17 | 41.67 | 38.61 | 29.67 | 31.67 | 34.13 |
| Qwen2.5-VL-7B | 53.33 | 64.44 | 70.00 | 57.50 | 73.33 | 49.00 | 35.00 | 59.78 |
| Qwen2.5-VL-72B | 48.89 | 61.11 | 75.00 | 52.50 | 78.89 | 40.00 | 31.67 | 57.83 |
| InternVideo2.5-Chat-8B | 34.66 | 36.87 | 54.17 | 37.87 | 48.44 | 27.03 | 30.51 | 38.51 |
| InternVL2.5-8B | 38.89 | 55.56 | 59.17 | 43.33 | 60.56 | 29.00 | 21.67 | 45.87 |
| InternVL2.5-26B | 45.56 | 48.89 | 60.83 | 44.17 | 63.89 | 29.67 | 20.00 | 46.88 |
| InternVL2.5-78B | 47.22 | 60.00 | 68.33 | 51.67 | 68.06 | 30.00 | 31.67 | 51.88 |
| MiniCPM-o 2.6 | 46.11 | 52.78 | 61.67 | 45.00 | 63.61 | 26.33 | 25.00 | 46.96 |
| LLaVA-NeXT-Video | 87.78 | 82.78 | 90.00 | 90.42 | **93.89** | 76.33 | 68.33 | 85.94 |
| + SafeVid-350K | +43.34 | +35.56 | +32.50 | +45.84 | +29.72 | +47.33 | +46.66 | +39.71 |
| Qwen2.5-VL-7B | **91.11** | **89.44** | 82.50 | **93.33** | 92.22 | **89.00** | 78.33 | **89.78** |
| + SafeVid-350K | +37.78 | +25.00 | +12.50 | +35.83 | +18.89 | +40.00 | +43.33 | +30.00 |

including those employing jailbreaking techniques, designed to probe deeper safety vulnerabilities. To further assess the generalization of safety alignment and provide a comprehensive evaluation beyond our curated data, we also evaluate models on a suite of established out-of-distribution (OOD) benchmarks. These are detailed in Table 1 and include Image-focused Safety (i.e., MM-SafetyBench [26], VLBreakBench [40], and miniJailBreakV-28K [28]) and Text-focused Safety (i.e., HarmEval [8] and StrongReject [36]). These OOD benchmarks help provide a more holistic view of the improvements achieved through SAFEVID.

**Evaluation Metrics.** For VidSafeBench, model performance is assessed based on Safety Rate across seven distinct harmful categories. Safety Rate quantifies the percentage of model responses that are deemed safe and do not violate safety guidelines. For the other benchmarks, we report Safety Rate and, where applicable, Helpful Rate as primary metrics. Helpful Rate measures the model's ability to provide useful and harmless responses to benign queries. This metric is particularly important for some earlier benchmarks where there's a potential for data leakage (i.e., the model might have seen questions during its pre-training or alignment). A high Safety Rate coupled with a low Helpful Rate might indicate that a model is not genuinely safer but has merely memorized to refuse prompts from the benchmark, rather than demonstrating nuanced safety understanding. Thus, Helpful Rate helps ensure that safety alignment does not unduly compromise utility or mask

Table 3: Out-of-Distribution (OOD) safety evaluation results. We report Safety Rate (%) and, where applicable, Helpful Rate (%) on established image-text and text-only safety benchmarks. VLBreakBench reports attack success rate (lower is better), while other metrics are higher is better.

| Model | Image + Text | | | | | | | Text | | |
|---|---|---|---|---|---|---|---|---|---|---|
| | VLBreakBench (↓) | | MMSafety-Bench (↑) | | | miniJailBreakV-28K (↑) | | HarmEval (↑) | StrongReject (↑) | |
| | Base | Challenge | SD | TYPO | SD+T | Safety | Helpful | Safety | Safety | Helpful |
| Claude-3.5-sonnet | **1.09** | **19.65** | **96.00** | 89.29 | 91.17 | **92.86** | 86.54 | **97.64** | 99.68 | 63.46 |
| GPT-4o | 8.52 | 46.31 | 93.58 | **95.95** | **92.78** | 85.00 | 17.65 | 96.36 | **100** | 1.28 |
| GPT-4o-mini | 14.84 | 72.21 | 84.39 | 86.80 | 82.59 | 80.00 | 24.55 | 92.55 | 99.68 | 1.60 |
| Gemini-2.0-flash | 53.38 | 66.84 | 78.37 | 74.33 | 68.87 | 36.07 | 92.05 | 89.49 | 8.33 | 15.38 |
| Gemini-2.0-flash-think | 20.63 | 71.44 | 77.11 | 70.26 | 67.48 | 30.11 | 97.62 | 89.27 | 7.02 | 0.00 |
| Gemini-1.5-pro | 26.53 | 64.94 | 80.87 | 80.82 | 73.19 | 37.14 | 99.04 | 90.91 | 1.28 | **100** |
| LLaVA-NeXT-Video | 68.00 | 63.84 | 62.57 | 49.19 | 42.02 | 30.36 | 90.59 | 74.73 | 0.96 | 66.67 |
| LLaVA-OneVision | 28.82 | 57.60 | 77.25 | 62.72 | 55.14 | 50.36 | 75.89 | 78.36 | 10.54 | 72.73 |
| Qwen2.5-VL-3B | 42.03 | 60.88 | 70.24 | 62.65 | 55.26 | 55.00 | 64.94 | 82.18 | 92.33 | 23.18 |
| Qwen2.5-VL-7B | 24.24 | 63.84 | 77.40 | 79.57 | 68.23 | 66.07 | 89.19 | 86.18 | 77.32 | 85.95 |
| Qwen2.5-VL-72B | 22.27 | 59.28 | 77.78 | 83.60 | 70.27 | 53.57 | 78.67 | 91.18 | 98.08 | 49.19 |
| InternVideo2.5-Chat-8B | 19.43 | 58.84 | 79.73 | 70.77 | 61.94 | 66.43 | 47.85 | 92.53 | 96.49 | 14.57 |
| InternVL2.5-8B | 24.67 | 61.36 | 80.92 | 73.91 | 65.66 | 72.50 | 62.56 | 92.18 | 94.89 | 35.35 |
| InternVL2.5-26B | 25.33 | 67.97 | 82.31 | 74.14 | 69.31 | 77.14 | 78.70 | 93.73 | 98.08 | 29.97 |
| InternVL2.5-78B | 19.65 | 63.76 | 84.20 | 77.90 | 69.18 | 72.14 | 66.83 | 96.91 | 99.68 | 14.42 |
| MiniCPM-o 2.6 | 59.72 | 65.57 | 71.43 | 52.41 | 48.68 | 39.29 | 91.82 | 86.18 | 4.15 | 76.92 |
| LLaVA-NeXT-Video | 16.48 | 43.57 | 73.40 | 68.98 | 74.31 | 51.07 | 97.90 | 94.00 | 22.68 | **100** |
| + SafeVid-350K | −51.52 | −20.27 | +10.83 | +19.79 | +32.29 | +20.71 | +7.31 | +19.27 | +21.72 | +33.33 |
| Qwen2.5-VL-7B | 3.93 | 28.67 | 88.31 | 90.77 | 81.93 | 91.79 | **100** | 94.55 | 99.68 | 99.04 |
| + SafeVid-350K | −20.31 | −35.17 | +10.91 | +11.20 | +13.70 | +25.72 | +10.81 | +8.37 | +22.36 | +13.09 |

over-cautious refusal patterns stemming from potential data memorization. Drawing inspiration from recent automated safety evaluation practices and to ensure a consistent, objective standard across all comparisons, we utilize GPT-4o to adjudicate these safety and helpfulness judgments.

## 4.2 Experimental Results

**Performance on SafeVidBench.** Table 2 details the safety performance of various VLMMs on SafeVidBench. On SafeVidBench-Base, which features everyday adversarial questions, unaligned SOTA models like LLaVA-NeXT-Video and Qwen2.5-VL-7B achieve average safety rates of 53.99% and 77.03%, respectively. This indicates inherent vulnerabilities to video-contextualized harmful queries. After alignment with SafeVid-350K dataset using DPO, LLaVA-NeXT-Video + SafeVid-350K achieves an impressive average safety rate of 96.38% on SafeVidBench-Base and 85.94% on SafeVidBench-Challenge. This represents a substantial improvement of 42.39% on the Base set and 39.71% on the Challenge set. Similarly, Qwen2.5-VL-7B + SafeVid-350K sees its average safety rate increase to 95.87% on Base and 89.78% on Challenge. The SafeVidBench-Challenge set, with its more subtle and human-red-teamed adversarial queries, presents a tougher evaluation. While all models score lower on this set, the SAFEVID aligned models consistently maintain significantly higher safety rates, underscoring the robustness imparted by our framework. Proprietary models like Claude-3.5-sonnet also exhibit strong baseline safety, setting high benchmarks, yet our aligned open-source models approach or even match these levels on specific categories.

**Out-of-Distribution (OOD) Generalization.** To evaluate whether the safety improvements generalize beyond our specific dataset, we test the models on a suite of established OOD benchmarks, as shown in Table 3. These include image-text safety benchmarks and text-only safety benchmarks. On VLBreakBench, where a lower attack success rate indicates better safety, LLaVA-NeXT-Video + SafeVid-350K reduces the success rate from 68.00% to 16.48% on the base set and from 63.84% to 43.57% on the challenge set. On MM-SafetyBench (SD+TYPO), the aligned LLaVA-NeXT-Video improves its safety score from 42.02% to 74.31%. For text-only benchmarks, such as HarmEval, the aligned LLaVA-NeXT-Video improves its safety rate from 74.73% to 94.00%. Notably, the Helpful Rate on benchmarks like miniJailBreakV-28K and StrongReject generally remains high or even improves for aligned models (e.g., LLaVA-NeXT-Video + SafeVid-350K achieves 100% Helpful Rate on StrongReject), indicating that our safety alignment does not unduly compromise the model's utility or lead to overly cautious refusals on benign OOD queries.

**Impact on General Capabilities (Alignment Tax).** A crucial consideration for any safety alignment method is its potential impact on the model's core capabilities, often referred to as the alignment tax. We investigate this by evaluating models on a general video question-answering benchmark (i.e., MMBench-Video [14]) that cover diverse perception and reasoning skills. The results in Figure 4 indicate that SAFEVID alignment incurs a minimal overall tax on these general video

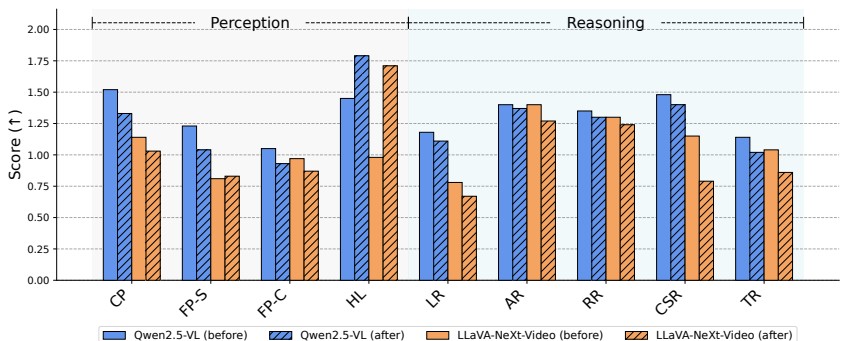

Figure 4: Impact of SAFEVID alignment on general VLMM capabilities, evaluated on MMBench-Video. Performance scores (higher is better) on perception (CP: Coarse Perception, FP-S: Fine-grained Perception [Single-Instance], FP-C: Fine-grained Perception [Cross-Instance], HL: Hallucination) and reasoning (LR: Logic Reasoning, AR: Attribute Reasoning, RR: Relation Reasoning, CSR: Common Sense Reasoning, TR: Temporal Reasoning) categories are shown for LLaVA-NeXT-Video and Qwen2.5-VL before and after alignment with SafeVid-350K.

understanding capabilities. For instance, LLaVA-NeXT-Video (after) sees its mean perception score slightly improve from 0.975 to 1.11, while its mean reasoning score shows a minor decrease from 1.14 to 0.99. Similarly, Qwen2.5-VL (after) exhibits modest drops in mean perception (1.3125 to 1.2725) and reasoning (1.31 to 1.24) scores. Notably, both models demonstrate significant improvements in the Hallucination (HL) category—an aspect of the Honest within our 3H framework. LLaVA-NeXT-Video's HL score increases from 0.98 to 1.71, and Qwen2.5-VL's improves from 1.45 to 1.79. This suggests that our framework can significantly enhance safety, particularly in promoting honest responses, without substantially degrading the model's overall utility.

**Data Scale.** To understand the relationship between the volume of preference data and alignment effectiveness, we conduct experiments varying the scale of data. Figure 3 illustrates these results, plotting model safety performance against the fraction of the full SafeVid-350K dataset utilized, ranging from 0.3% to 100%. The results demonstrate a clear positive correlation between data scale and safety improvements. On SafeVidBench-Base, significant gains in Safety Rate are observed even with relatively small fractions of the data (e.g., 10-20%), though performance continues to climb as more

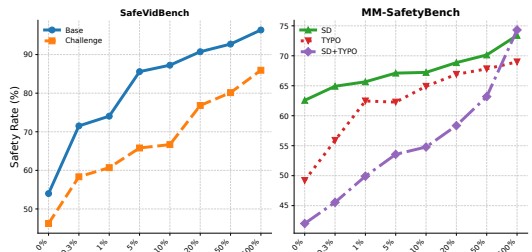

Figure 3: Impact of SafeVid-350K data scale on alignment effectiveness. Safety Rate is evaluated on SafeVidBench and MM-SafetyBench.

data is added. Notably, performance on the more difficult SafeVidBench-Challenge set and the OOD MM-SafetyBench benchmark shows a steeper improvement curve and benefits more substantially from larger data fractions. This suggests that while foundational safety refusals can be learned with moderate data, achieving robustness against sophisticated attacks and generalizing safety principles to related domains requires more comprehensive preference supervision.

**Human Validation.** To address potential concerns regarding the quality of our generated data and the reliability of our automated evaluation, we conduct a manual verification study. First, we randomly sample 1,000 preference pairs from SafeVid-350K dataset. Three expert annotators are tasked to assess whether the chosen response is indeed safer than the rejected response. The human evaluation confirms the synthetic preference in 97.21% of the cases, demonstrating the high quality and alignment of our dataset generation pipeline. Second, to validate the accuracy of using GPT-4o as an automated judge, we sample 500 model outputs from both the Base and Challenge sets. The judgments from our human experts align with GPT-4o's safety verdicts in 92.54% of the instances. This high level of agreement substantiates the reliability of our automated evaluation process.

**Limitations.** While SAFEVID demonstrably improves VLMMs' safety, the framework's reliance on textual video descriptions as an interpretive bridge means its efficacy is linked to the fidelity of these textual proxies. Consequently, highly subtle visual-temporal safety nuances that are exceptionally challenging to capture exhaustively in text, or rapidly evolving misuse patterns not yet

fully encapsulated by our current safety taxonomy, may still present nuanced edge cases, suggesting avenues for ongoing refinement of both the descriptive granularity and the scope of safety principles.

## 5 Conclusion

In this work, we introduce SAFEVID, a novel framework aimed at mitigating safety risks specific to Video Large Multimodal Models (VLMMs). SAFEVID leverages detailed textual narratives of video content as an intermediary, enabling the application of established text-based safety reasoning to the complex domain of video understanding. The framework comprises three key components: the construction of *SafeVid-350K*, a large-scale preference dataset focused on video scenarios; the use of Direct Preference Optimization for targeted safety alignment; and the development of a comprehensive safety benchmark, *SafeVidBench*. Experimental results demonstrate the effectiveness of SAFEVID, showing substantial improvements in VLMM safety compliance and highlighting the promise of language-based representations for instilling safety principles in video modality.

## Acknowledgments

This work is in part supported by National Key R&D Program of China (Grant No. 2022ZD0160103) and National Natural Science Foundation of China (Grant No. 62276067), and Shanghai Artificial Intelligence Laboratory.

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
